# Incidence, risk factors and outcomes of preterm live births in a tertiary health facility in Ghana

Anthony Kwame Dah[1,2☯], Joseph Osarfo[3☯]*, Hintermann Mbroh[2], Gifty Dufie Ampofo[3], Adu Appiah-Kubi[1,2], Wisdom Klutse Azanu[1,2], Michael Amoh[1,2], Emmanuel Senanu Komla Morhe[1,2]

1 Department of Obstetrics and Gynaecology, School of Medicine, University of Health and Allied Sciences, Ho, Volta Region, Ghana, 2 Department of Obstetrics and Gynaecology, Ho Teaching Hospital, Ho, Volta Region, Ghana, 3 Department of Community Health, School of Medicine, University of Health and Allied Sciences, Ho, Volta Region, Ghana

☯ These authors contributed equally to the work and are joint first authors.
* josarfo@uhas.edu.gh

## Abstract

### Objectives

Accurate measures of the burden of preterm live births are a challenge, particularly in low-and-middle income countries, due to factors including poor data quality and unreliable methods using last menstrual period (LMP) in determining gestational age of pregnancies. This study employed gestational age from first trimester ultrasound to determine incidence of preterm livebirths at Ho Teaching Hospital in Ghana, their risk factors and adverse outcomes.

### Methods

This was a prospective study involving 666 pregnant women and their 680 live newborns from 1st October, 2019–31st March, 2020. Data was collected on socio-demographic characteristics such as age and for other variables including gestational age at delivery and birthweight. The primary outcome was overall preterm livebirth incidence. Logistic regression analysis was used to determine its predictors. Odds ratios were reported with 95% confidence intervals. Statistical significance was pegged at p-value <0.05.

### Results

The mean age of the women was 29.0 years. About 93% (616/666) had formal education. Overall preterm livebirth incidence was 9.1% (62/680). Formal employment [AOR 6.84 (95% CI 1.63–28.73), p = 0.009], informal employment [AOR 4.98 (95% CI 1.39–17.85), p = 0.014] and hypertensive disorders in pregnancy (HDP) [AOR 6.43(95% CI 2.78–14.89); p < 0.001] increased the odds of preterm livebirth. Adequate antenatal clinic contacts (ANC) [AOR 0.14 (95% CI 0.05–0.37), p < 0.001]

**Data availability statement:** All relevant data are within the manuscript and its Supporting Information files.

**Funding:** The author(s) received no specific funding for this work.

**Competing interests:** The authors have declared that no competing interests exist.

and doses of sulphadoxine-pyrimethamine for malaria prevention in pregnancy (IPTp) [AOR 0.21(0.10–0.45), p < 0.001] reduced the odds of preterm livebirth by 86% and 79% respectively.

## Conclusion

Overall preterm livebirth incidence in the study compared favourably with global estimates. HDP and being employed were important risk factors and. suggest reducing workloads of pregnant women in formal employment may be beneficial. Targeted posting of specialists to aid adequate management of HDP at district hospitals and mitigating missed opportunities for IPTp need to be considered.

## Introduction

Preterm birth refers to delivery before gestational age of 37 completed weeks and poses increased risks of morbidity and mortality for the newborn. Worldwide, the burden of preterm live births remains high and occurred in at least 1 in 10 live births by 2015 [1]. There are conflicting reports on the global trend of preterm live births. While some reviews show increments from 9.8% in 2000 to 10.6% in 2014 [2], others variously report a reduction by 5.3% from 16.1 million in 1990 to 15.2 million 2019 [3] or no significant change in preterm live births worldwide from 2010 to 2020 [4]. Asia and sub-Saharan Africa contribute 81.1% of the global burden [1]. Despite the reported data, obtaining accurate measures of the burden of preterm births, especially in low- and middle-income countries (LMICs), is challenging [5,6]. This mostly arises from data quality issues including the use of less accurate methods such as last menstrual period (LMP) in dating pregnancies [5,6] even though ultrasonography is increasingly available. This presents a gap in understanding the effective strategies to reduce the rates of preterm live births.

Preterm live babies are vulnerable to adverse outcomes including low birth weight with its sequelae, low Apgar scores, sepsis, jaundice, anaemia, early neonatal deaths, neurological and cognitive developmental deficits, increased risk of neonatal intensive care unit (NICU) admissions and chronic health problems such as psychiatric, respiratory, endocrine/metabolic, cardiovascular, and renal disorders [6,7]. These outcomes are disproportionately higher in low- and middle-income countries including Ghana [6,8,9]. Extremes of maternal ages (< 20 years and > 35 years), hypertensive disorders in pregnancy, inadequate antenatal care clinic (ANC) contacts, and premature rupture of membranes (PROM) are commonly reported determinants of preterm live births in low- and middle-income countries [5,10,11]. In Ethiopia, women with <4 ANC contacts and PROM had about 5- and 12-times increased odds of preterm live births respectively though the confidence intervals around the adjusted odds ratios were appreciably wide [12]. Other risk factors such as maternal illiteracy and nulliparity, HIV infection, urinary tract infection, anaemia during pregnancy, multiple pregnancy and rural residence, previous history of preterm birth, short inter-pregnancy interval and malaria have been reported [5,7,9,11,13,14].

Literature review did not show any current national level data on preterm live birth rate for Ghana. However, individual studies report a wide range of occurrence, 9%−70%, of preterm deliveries [15–20]. Agbeno et al [16] reported a 4.7% 10-year prevalence in Cape Coast Teaching Hospital. Only the reports by Otieku et al [18] and Anto et al [20] were prospective among these studies. While the study by Otieku et al [18] did not indicate how gestational age was determined, that by Anto et al [20] employed a mix of LMP and first-trimester ultrasound.

Particularly in the Volta Region of Ghana, Axame et al [19] reported a 14.1% prevalence of preterm deliveries in the HTH in a retrospective study. Aside from the risk of poor data quality from relying on routine health services data as is often the case in retrospective designs, the study [19] also excluded multiple births and thus increased the potential for underestimation. Furthermore, the study by Axame et al [19] was conducted at a time when LMP was the prevalent method of gestational age determination. The HTH was designated a teaching hospital in April 2019 and has since adopted first-trimester USG for gestational age assessment. Against this background, there is a need to re-assess the burden of preterm births in the HTH in a prospective study that assures adequate data quality, solely employs USG for gestational age determination and includes both singleton and multiple childbirths. The present study reports the incidence, determinants and outcomes of preterm live births as part of a larger prospective study conducted to describe the incidence of stillbirths and preterm live births in HTH. The results pertaining to stillbirths have been published earlier [21].

## Methods

### Study site description, study design and study population

This was a prospective cohort study conducted at the Obstetrics and Gynaecology department of the HTH. Data collection was from 1st October, 2019–31st March, 2020. The study site has been described in detail in a previous study [21]. Briefly, HTH is the prime referral facility in the Volta Region and services parts of the neighbouring Eastern region and the republic of Togo. The obstetrics and gynaecology wing has maternity and labour wards, obstetric theatre and a neonatal intensive care unit (NICU). Antenatal clinic attendance was 6000 in 2021 and about 2000 deliveries are conducted annually (Biostatistics Unit, HTH, 2022). The study population were women admitted for labour and delivery or pre-labour caesarean section and their newborn babies at HTH within the study period.

### Sample size determination

No formal sample size calculation was done. All women admitted for labour and delivery or pre-labour caesarean sections from 1st October, 2019–31st March, 2020 and their newborns were included if found eligible but excluded those who had stillbirths or delivery before 28 weeks. Delivery before 28 weeks is considered an abortion in our settings. In all, 666 mothers and 680 live newborns from a total sampling of eligible participants over the defined period were included for data analysis.

### Study procedures and data collection tool

A pretested structured questionnaire was administered to eligible women to collect data on sociodemographic characteristics such as age and level of education, obstetric history of the index pregnancy including medical conditions such as hypertensive disorders in pregnancy, and number of doses of sulphadoxine-pyrimethamine for intermittent preventive treatment of malaria in pregnancy (IPTP-SP) and ANC contacts. Specifically, the questionnaires were administered when the women were admitted for labour or at 1–2 hours after delivery when they were stable and at admission for pre-labour caesarean section. Where a woman was in severe labour pain at admission, questionnaire administration was delayed till after delivery when she was stable. The questionnaires were administered by three registered midwives trained over two days in the relevant data collection procedures. Their training also included translation of the questionnaire from English to the local Ewe and Twi languages for uniformity as described in an earlier study [21]. Questionnaire administration was done in English, Ewe or Twi.

Except for five referred cases (less than 1% of the study women), gestational age at delivery was ascertained based on the estimated gestational age at ultrasound in the first trimester documented in the maternal and child health record book (antenatal book). Birth outcomes such as birthweight, Apgar scores, congenital anomaly and NICU admission were recorded at delivery or later extracted from the delivery register. The newborns that were discharged before 7th day of life were followed up at first postnatal clinic or with phone calls to obtain information on their life status a week after birth. In case of multiple pregnancies, data was collected on each birth separately with a unique number. For quality control, 60% of data collected were cross-checked with primary sources such as delivery registers by two residents in the obstetrics and gynaecology department. The choice of 60% was discretionary and involved cross-checking three out of every five questionnaires. The registered midwives used for data collection, by default, were already familiar with the study variables and were positioned to identify potential data discrepancies themselves. The choice of using 60% of data collected in quality control was thus deemed reasonable.

## Definition of variables

Preterm live birth was a delivery from gestational age of 28 weeks 0 days (28W0D) to 36W6D (referred to as 28–36) with spontaneous breath or heartbeat at birth.

Very early preterm live birth was a delivery from gestational age of 28W0D to 31W6D (referred to as 28–31) with spontaneous breath or heartbeat.

Early preterm live birth was a delivery from gestational age of 32W0D to 34W6D (referred to as 32–34) with spontaneous breath or heartbeat at birth.

Late preterm birth was a delivery from gestational age of 35W0D to 36W6D (referred to as 35–36) with spontaneous breath or heartbeat at birth.

Term live birth referred to a delivery at gestational age of 37W0D or more with spontaneous breath or heartbeat at birth.

Index pregnancy was the pregnancy and its birth outcomes' information extracted from the data source.

ANC contacts of four or more was labelled as adequate and three or less as inadequate.

IPTp dose during pregnancy of three or more was referred to as adequate and two or less as inadequate.

Hypertensive disorders in pregnancy included gestational hypertension, pre-eclampsia, eclampsia, chronic hypertension in pregnancy and chronic hypertension with superimposed pre-eclampsia.

Anaemia in pregnancy was the last haemoglobin check of less than 11g/dl within one month prior to delivery.

Low birth weight (LBW) was a neonate born after 28 completed weeks' gestation with weight less than 2.5 kg at birth (less than 1 kg was extremely LBW, 1 kg – 1.499 kg very LBW and 1.5 kg – 2.499 kg as moderately LBW.

Normal birth weight was a weight of a neonate born at 28W0D gestation or more with a birth weight of 2.5 kg or more.

Non-assuring 5- minute Apgar score referred to Apgar score of 6 or less at 5th minute of birth (consisting 6−4 as moderately low and 3 or less as low 5-minute Apgar scores).

Early neonatal death referred to the study neonate that died within 7 days after birth.

## Data management and analysis

The filled questionnaires were checked for completeness and accuracy. The data was double entered in Microsoft Excel and exported into SPSS version 22 (IBM Corp, Armonk, NY, USA) for cleaning, coding and analysis. The primary study outcome was overall preterm live birth incidence (incidence proportion). The secondary outcomes were the incidence (incidence proportion) of very early preterm live birth, early preterm live birth, late preterm live birth, adverse preterm live birth outcomes including low birth weight, non-assuring 5-minute Apgar scores, NICU admissions and early neonatal deaths and predictors of overall preterm live birth. Independent variables included socio-demographic characteristics such as age, educational level, employment and residence of the women, antenatal attendance and medical conditions

during index pregnancy including hypertensive disorders and anaemia in pregnancy. Descriptive analyses were done and presented as frequencies, percentages, means and standard deviation. Associations between preterm live births and the independent variables were assessed using logistic regression methods. Variables that showed p-values of 0.1 or less in univariate analysis were entered in a multivariate regression. Crude and adjusted odds ratios were reported with 95% confidence intervals. For the final multivariate model, statistical significance was pegged at $p < 0.05$.

### Ethical considerations

The University of Health and Allied Sciences Research and Ethics Committee granted approval for the study (protocol ID UHAS-REC A12 [7] 18–19) and permission to collect data was given by the management of HTH. Written informed consent was obtained from the study participants. Women who were illiterate thumb printed the consent forms after the study objectives were explained to them in a language they understood and in the presence of a witness (their relative). Participants under 18 years gave assent and written informed consent was obtained from their caregivers/guardians after explaining the purpose of the study to them. It was explained that participation was voluntary and that refusal to participate would not lead to denial of care or any other punitive measure. They were assured of confidentiality. Study identification numbers were used to anonymize the participants.

## Results

### Background characteristics of study participants

Data on 666 mothers and 680 live newborns were analyzed and reported. The background characteristics of the study participants are summarized in Table 1. The women's ages ranged from 13–46 years with the mean age (SD) being 29 years (6.28). About 71% (477) of the study women were in the age group 20–34 years. More than half, 55.3% (368) of the mothers had had either senior high school or tertiary education. Majority, 92.5% (616), lived in urban/peri-urban areas during the index pregnancy period, three-quarters (489) were married, and 33.5% (228) had no previous delivery experiences.

The obstetric characteristics of the study women are presented in Table 2. Six hundred and two women (90.3%) had adequate ANC contacts and 70.6% (470) took adequate doses of IPTp. Hypertensive disorders in pregnancy occurred in about a tenth, 8.9% (59), and premature rupture of membranes (PROM) in 5.7% (38) of the women. About half, 51.5% (343), of the women had anemia and 2.3% (15) were sickle cell patients. One hundred and thirty-two participants (19.8%) were referred from catchment health facilities for delivery at HTH during the study period. Labour was induced in 9.5% (63) of the study women.

### Preterm live births and adverse outcomes

Out of 680 live births within the study period, preterm births occurred in 9.1% (62); comprising 38.7% (24) very early preterm, 25.8% (16) early preterm and 35.5% (22) late preterm births (see Table 3).

Low birth weight occurred in 12.5% (85) of all the live newborns. Of the 62 preterm live births 91.9% (57) had low birth-weight; comprising three extremely low birth weight (2 among very early preterm and 1 early preterm births), 26.3% (15) very low birth weight and 62.9% (39) moderately low birth weight.

Thirty-eight (5.8%) newborns suffered non-assuring 5-minute Apgar scores. Three (50%) of newborns with low 5-minute Apgar scores were preterm, and 63.6% of moderately low 5-minute Apgar scores among preterm births occurred in very early preterm newborns.

Over the study period, 42.9% (51) of the 119 (17.5%) admissions to NICU from the study were preterm births. Of these, 41.1% (21) constituted very early preterm births (see Table 3). Four early neonatal deaths occurred among preterm newborns and all four were very early preterm birth neonates.

**Table 1. Demographic characteristics of the study women.**

| Variable | Frequency (N = 666) | Percent (%) |
|---|---|---|
| **Age in years** | | |
| <20 | 52 | 7.8 |
| 20 - 34 | 477 | 71.4 |
| >35 | 137 | 20.6 |
| **Formal education** | | |
| None | 50 | 7.5 |
| Basic school | 248 | 37.2 |
| SHS | 139 | 20.9 |
| Tertiary | 229 | 34.4 |
| **Employment** | | |
| Unemployed | 114 | 17.1 |
| Informal | 349 | 52.4 |
| Formal | 203 | 30.5 |
| **Marital status** | | |
| Single | 165 | 24.8 |
| Married | 489 | 73.4 |
| No response | 12 | 1.8 |
| **Residence** | | |
| Rural | 50 | 7.5 |
| Urban/peri-urban | 616 | 92.5 |
| **Previous delivery** | | |
| Zero | 228 | 33.5 |
| 1 - 3 | 407 | 59.9 |
| >4 | 35 | 5.1 |
| No response | 10 | 1.5 |

### Risk factors of preterm live birth

In the bivariate analyses, informal employment doubled the odds of preterm live births [OR 2.15(95% CI 1.10–4.18); p = 0.024] compared to unemployment (see Table 4). Women with hypertensive disorders in pregnancy had 8 times the odds of preterm delivery [OR 8.17(95% CI 4.42–15.07); p < 0.001] while those with PROM showed about 4 times the odds [OR 3.77(95% CI 1.75–8.14); p = 0.001] compared to parturient without these two obstetric complications. Also, referred women with complicated pregnancies from peripheral health facilities for delivery at HTH showed 4 folds [OR 3.98 995% CI 2.31–6.84), p < 0.001] increase in the odds of preterm live births compared to the women who attended ANC at HTH. On the contrary, singleton pregnancy [OR 0.07 (95% CI 0.03–0.16); p < 0.001], adequate ANC contacts [OR 0.20 (95% CI 0.10–0.40); p < 0.001] and adequate IPTp doses [OR 0.20 (95% CI 0.12–0.36); p < 0.001] reduced the odds of preterm live births by 93%, 80% and 80% respectively compared to women with twin pregnancy, inadequate ANC contacts and inadequate IPTp doses.

In the multivariate analysis (see Table 4), formal employment [AOR 6.84 (95% CI 1.63–28.73), P = 0.009] and informal employment [AOR 4.98 (95% CI 1.39–17.85), p = 0.014] were found to increase the odds of preterm live birth by 7 and 5 folds respectively compared to unemployed women. The odds of preterm live birth for hypertensive disorders in pregnancy [AOR 6.43 (95% CI 2.78–14.89), p < 0.001] and referral for delivery at HTH [AOR 2.65 (95% CI 1.24–5.66), p = 0.012] remained elevated. Those for singleton pregnancy [AOR 0.05 (95% CI 0.02–0.13); p < 0.001], adequate ANC

**Table 2. Obstetric factors of index pregnancy and mode of delivery within the study period.**

| Variable | Frequency (N = 666) | Percent (%) |
|---|---|---|
| **ANC contact** | | |
| 0 | 2 | 0.3 |
| 1 - 3 | 47 | 7.1 |
| ≥4 | 602 | 90.3 |
| No response | 15 | 2.3 |
| **IPTp dose** | | |
| Zero | 22 | 3.3 |
| 1 - 2 | 128 | 19.2 |
| ≥3 | 470 | 70.6 |
| No response | 46 | 6.9 |
| **Fetus(es) in a pregnancy** | | |
| $Twin | 15 | 2.3 |
| Singleton | 651 | 97.7 |
| **\*Hypertension** | | |
| Yes | 59 | 8.9 |
| No | 607 | 91.1 |
| **Diabetes in pregnancy** | | |
| Yes | 13 | 2.0 |
| No | 648 | 97.3 |
| No response | 5 | 0.8 |
| **&Anaemia in pregnancy** | | |
| Yes | 343 | 51.5 |
| No | 309 | 46.4 |
| No response | 14 | 2.1 |
| **@SCD in pregnancy** | | |
| Yes | 15 | 2.3 |
| No | 600 | 90.1 |
| No response | 51 | 7.7 |
| **PROM** | | |
| Yes | 38 | 5.7 |
| No | 628 | 94.3 |
| **Referral status** | | |
| Referred | 132 | 19.8 |
| Not referred | 534 | 80.2 |
| **Labour** | | |
| Induced | 63 | 9.5 |
| Not induced | 603 | 90.5 |
| **Mode of delivery (N = 680)** | | |
| Caesarean | 215 | 31.6 |
| Vaginal | 465 | 68.4 |

*Hypertension – hypertension disorders in pregnancy @SCD in pregnancy – sickle cell disease in pregnancy &Mild anaemia (10.9 – 9.1g/dl) 63.0%, moderate anaemia (9.0–7.1g/dl) 33.5% and severe anaemia (≤ 7.0g/dl) 3.5% $One twin newborn was stillbirth and excluded from analysis. This explains why the newborns are 680 instead of 681.

**Table 3. Live births of index pregnancy and resultant newborns characteristics.**

| Outcome variables | Live births (gestational age in completed weeks) N(%) | | | | |
| --- | --- | --- | --- | --- | --- |
| | Preterm live birth subtypes | | | Preterm live births | Term live birth |
| | 28 - 31 | 32 - 34 | 35 − 35 | 28 - 36 | ≥ 37 |
| **Birth weight in kg** | | | | | |
| < 1.000 | 2(66.7) | 1(33.3) | 0 | 3(100) | 0 |
| 1.000–1.499 | 10(66.6) | 4(26.7) | 1(6.7) | 15(75.0) | 5(25.0) |
| 1.500–2.499 | 11(28.2) | 11(28.2) | 17(43.6) | 39(62.9) | 23(37.1) |
| ≥ 2.500 | 1(20.0) | 0 | 4(80.0) | 5(0.8) | 590(99.2) |
| **5-minute Apgar score** | | | | | |
| 0 - 3 | 1(33.3) | 2(66.7) | 0 | 3(50.0) | 3(50.0) |
| 4 - 6 | 7(63.6) | 2(18.2) | 2(18.2) | 11(34.4) | 21(65.6) |
| ≥ 7 | 16(33.3) | 12(25.0) | 20(41.7) | 48(7.5) | 594(92.5) |
| **NICU admission** | | | | | |
| Admitted | 21(41.1) | 14(27.5) | 16(31.4) | 51(42.9) | 68(57.1) |
| Not admitted | 3(27.3) | 2(18.2) | 6(54.5) | 11(2.0) | 550(98.0) |
| **Status at day 7 after birth** | | | | | |
| Died | 4(100) | 0 | 0 | 4(57.1) | 3(42.9) |
| Alive | 20(34.5) | 16(27.6) | 22(37.9) | 58(8.6) | 615(91.4) |
| **Total** | 24(38.7) | 16(25.8) | 22(35.5) | 62(9.1) | 618(90.9) |

contacts [AOR 0.14 (95% CI 0.05–0.37); p < 0.001] and adequate IPTp dose administration during index pregnancy [AOR 0.21(95% CI 0.10–0.45); p < 0.001] also remained significantly reduced (see Table 4).

## Discussion

This study aimed at providing a reliable information on the incidence, determinants and outcomes of preterm live births in a prospective study utilizing first trimester USG for gestational age determination at Ho Teaching Hospital. An overall preterm live birth incidence of 9.1%, comprising 38.7% very early preterm live births, 25.8% early preterm live births and 35.5% late preterm live births, was found. Employment, referral for delivery at HTH and hypertensive disorders in pregnancy were identified as the risk factors for preterm live births while singleton pregnancy, adequate ANC contacts and adequate IPTp dose administration during pregnancy were protective. Preterm birth adverse outcomes, particularly low 5-minute Apgar scores and early neonatal deaths, were dominant among the very early preterms.

The 9.1% preterm live birth incidence in the current study is comparable with previous estimates of 8.7%−9.9% globally and in the Euro-Asian continents [2,4,8,11]. It falls within the range of 3.4% to 49.4% reported for sub-Saharan Africa in a scoping review [5]. Comparing to Ghanaian studies, the present study's preterm livebirths incidence is lower than the 12.5%−37.3% reported in Komfo Anokye and Korle Bu teaching hospitals [10,19] but comparable to the 9.0% in Cape Coast teaching hospital over 2010–2019 [16]. A previous study at the study site reported a 14.1% prevalence of preterm live births compared to the 9.1% incidence in the current study [19].

The challenges preterm live births studies face in LMICs such as limited resources and poor data quality particularly in retrospective studies [6], coupled with common use of LMP for calculating gestational age of pregnancies [5] could partly account for the wide variations in prevalence of preterm live birth observed across and within same facilities in Ghana. For instance, an earlier study reporting preterm live birth prevalence at HTH [19] was a retrospective review of facility records from October to December 2018 and excluded multiple pregnancy and newborns with congenital anomalies. In contrast, the current study was based on primary data collected prospectively between October 2019 and March 2020 when first

**Table 4. Bivariate and multivariate logistic regression analyses output for association between preterm live birth and independent variables.**

| Predictor Variable | Outcome Variable | | Crude OR | 95% CI | p-value | AOR | 95% CI | p-value |
|---|---|---|---|---|---|---|---|---|
| | Preterm n(%) | Term n(%) | | | | | | |
| **Age group (years)** | | | | | | | | |
| <20 | 5(9.4) | 48 (90.6) | 0.94 | 0.32-2.74 | 0.906 | | | |
| 20 - 34 | 43 (8.8) | 444(91.2) | 1 | | | | | |
| ≥35 | 14(10.0) | 126(90.0) | 0.87 | 0.46-1.64 | 0.671 | | | |
| **Formal education** | | | | | | | | |
| No | 6(12.0) | 44(88.0) | 1.86 | 0.69-5.01 | 0.221 | | | |
| Basic | 25(9.8) | 230(90.2) | 1.48 | 0.77-2.85 | 0.239 | | | |
| SHS | 15(10.6) | 126(89.4) | 1.62 | 0.78-3.39 | 0.199 | | | |
| Tertiary | 16(6.8) | 218(93.2) | 1 | | | | | |
| **Employment** | | | | | | | | |
| Formal | 12(5.8) | 194(94.2) | 0.84 | 0.33-2.11 | 0.703 | 6.84 | 1.63-28.73 | 0.009 |
| Informal | 42(11.7) | 316(88.3) | 1.80 | 0.82-3.94 | 0.045 | 4.98 | 1.39-17.85 | 0.014 |
| Unemployment | 8(6.9) | 108(93.1) | 1 | | | 1 | | |
| **Marital status** | | | | | | | | |
| Single | 21(12.4) | 148(87.6) | 1.77 | 1.01-3.12 | 0.048 | | | |
| Married | 37(7.4) | 462(92.6) | 1 | | | | | |
| **Residence** | | | | | | | | |
| Rural | 7(13.7) | 44(86.3) | 1.66 | 0.71-3.86 | 0.239 | | | |
| Urban/peri-urban | 55(8.7) | 574(91.3) | 1 | | | | | |
| **Previous delivery** | | | | | | | | |
| Zero | 25(11.0) | 203(89.0) | 2.03 | 0.46-8.99 | 0.350 | | | |
| 1 - 3 | 32(7.9) | 374(91.9) | 1.41 | 0.32-6.14 | 0.649 | | | |
| ≥4 | 2(5.7) | 33(94.3) | 1 | | | | | |
| **Number of Fetus in pregnancy** | | | | | | | | |
| Twin | 15(51.7) | 14(48.3) | 1 | | | 1 | | |
| Singleton | 47(7.2) | 604(92.8) | 0.07 | 0.03-0.16 | <0.001 | 0.05 | 0.02-0.13 | <0.001 |
| **ANC contact** | | | | | | | | |
| ≤3 | 14(28.0) | 36(72.0) | 1 | | | 1 | | |
| ≥4 | 44(7.2) | 569(92.8) | 0.20 | 0.10-0.40 | <0.001 | 0.14 | 0.05-0.37 | <0.001 |
| IPTp | | | | | | | | |
| ≤2 | 32(20.6) | 123(79.4) | 1 | | | 1 | | |
| ≥3 | 24(5.0) | 454(95.0) | 0.20 | 0.12-0.36 | <0.001 | 0.21 | 0.10-0.45 | <0.001 |
| ***Hypertension** | | | | | | | | |
| Yes | 22(36.1) | 39(63.9) | 8.17 | 4.42-15.07 | <0.001 | 6.43 | 2.78-14.89 | <0.001 |
| No | 40(6.5) | 579(93.5) | 1 | | | 1 | | |
| **Diabetes in pregnancy** | | | | | | | | |
| Yes | 2(15.4) | 11(84.6) | 1.82 | 0.40-8.42 | 0.441 | | | |
| No | 60(9.1) | 602(90.9) | 1 | | | | | |
| **Anaemia in pregnancy** | | | | | | | | |
| Yes | 29(8.4) | 318(91.6) | 0.95 | 0.55-1.63 | 0.837 | | | |
| No | 28(8.8) | 290(91.2) | 1 | | | | | |
| **&SCD in pregnancy** | | | | | | | | |
| Yes | 2(13.3) | 13(86.7) | 1.70 | 0.37-7.72 | 0.495 | | | |
| No | 51(8.3) | 561(91.7) | 1 | | | | | |

*(Continued)*

**Table 4.** (Continued)

| Predictor Variable | Outcome Variable | | Crude OR | 95% CI | p-value | AOR | 95% CI | p-value |
|---|---|---|---|---|---|---|---|---|
| | Preterm n(%) | Term n(%) | | | | | | |
| **PROM** | | | | | | | | |
| Yes | 10(25.0) | 30(75.0) | 3.77 | 1.75-8.14 | 0.001 | 2.78 | 0.88-8.83 | 0.083 |
| No | 52(8.1) | 588(91.9) | 1 | | | 1 | | |
| **Referral status** | | | | | | | | |
| Referred | 28(20.9) | 106(79.1) | 3.98 | 2.31-6.84 | <0.001 | 2.65 | 1.24-5.66 | 0.012 |
| Not referred | 34(6.2) | 512(93.8) | 1 | | | 1 | | |
| **Labour** | | | | | | | | |
| Induced | 3(4.8) | 60(95.2) | 0.47 | 0.14-1.56 | 0.217 | | | |
| Not induced | 59(9.6) | 558(90.4) | 1 | | | | | |
| **Mode of delivery** | | | | | | | | |
| Caesarean | 33(14.9) | 183(85.1) | 2.54 | 1.50-4.30 | 0.001 | 2.00 | 0.94-4.23 | 0.071 |
| Vaginal | 30(6.5) | 435(93.5) | 1 | | | 1 | | |

**\***Hypertension – Hypertensive disorders in pregnancy &Sickle Cell Disease

trimester ultrasound became routine in determining gestational age of pregnancies. Besides, multiple pregnancy and neonates with congenital anomaly were included in the present study. Thus, the variability in preterm live birth occurrence (14.1% vs 9.1%) in the two studies at HTH may partly stem from the change in practice from use of LMP to application of first trimester ultrasound in determining gestational age of pregnancy and differences in study designs and study population.

Similar to the current study's findings, hypertensive disorders in pregnancy are a frequently reported risk factor for preterm live births, especially in LMICs [5,8,17,22–27]. A previous Ghanaian study identified pre-eclampsia and eclampsia as the main risk factor for preterm live births [22]. Pre-eclampsia and eclampsia in preterm pregnancies are the most frequent referrals from the catchment health facilities for delivery at HTH due to their lack of a NICU facility and they often end up in preterm births. This could explain the observed association between referrals for delivery at HTH and preterm live births. Posting more obstetrics and gynaecology specialists to district hospitals to run specialized clinics for pregnancies at risk of preterm live births could contribute to reducing preterm live births burden in Volta Region, and Ghana as a whole [28].

The current study found formal and informal employment were risk factors for preterm live births and these align with studies in Uganda [29,30] and Indonesia where housewives had reduced odds of preterm live births compared to working women [31]. It however contradicts a study in Nigeria that found no association between maternal occupation and preterm births [32]. The employed pregnant woman is presumptively less likely to have enough time for adequate ANC contacts or seek healthcare early enough when needed. Besides, the employed pregnant woman, particularly in the informal sector, is likely to engage in strenuous activities with inadequate rest compared to the unemployed pregnant woman [33]. These findings suggest it may be useful to lessen workload and time spent at work for pregnant women. Revisions to the employment policy for reduced workloads for pregnant women in the formal sector of the economy could be considered.

Preterm live birth risks imposed by twin pregnancy, inadequate ANC contacts and inadequate IPTp dose administration during pregnancy are well documented in literature [11,20,27,34]. The current study observed a 95% reduction in the odds of preterm live births in singleton pregnancy compared to twin pregnancy. This finding buttresses the calls by many, including International Federation of Gynaecology and Obstetrics (FIGO), for a single embryo transfer in in-vitro fertilization to improve pregnancy outcomes [35].

In the present study, adequate ANC contacts and IPTp doses reduced the odds of preterm livebirths by 86% and 79% respectively. Aside from its primary role in malaria prevention, the sulphadoxine-pyrimethamine used for IPTp has anti-bacterial activity or prevents bacterial infections such as urinary tract and sexually transmitted infections which have been observed to be associated with preterm births [36]. Other studies, however, showed that the number of IPTp doses did not impact on gestational age at delivery [37–39]. Adequate ANC contacts increased tremendously in Ghana from 49.3% to 85.6% between 2006 and 2014 [13,40] and was at least 90% among parturients in the present study. Proportions of pregnant women receiving at least 3 doses of IPTp in Ghana, on the other hand, are lower and increased from 41% in 2012 to 55% in 2021 with only 17% receiving five doses also in 2021 [41]. The disparity between adequate ANC contacts and adequate IPTp doses in the current study (90.3% vs 71% respectively) suggests missed opportunities for delivering IPTp to pregnant women. This may arise from factors including stock-outs of sulphadoxine-pyrimethamine and low health worker experience [42]. Thus, the quality of ANC services including eliminating missing opportunities for IPTp also needs to be improved if the full benefit to preterm live births is to be realized.

## Strengths and limitations

The strength of the study lies in the fact that it was a prospective study which also employed first trimester ultrasound for the determination of gestational age of pregnancies unlike most previous studies that relied on retrospective study designs and the use of LMP [5,6,36]. This brings to bear an improvement in the quality of data collected for gestational age of pregnancy. The study was conducted at a single site and this could limit its generalizability. The Ho Teaching Hospital receives referrals from the entire Volta Region and beyond and thus the study results can be representative of the reality in the region. Also, the 95% confidence intervals around the AOR for hypertension and employment in the multivariate analysis were wide and should be interpreted with caution.

## Conclusion

The study reports an overall preterm live birth incidence of 9.1% at HTH with employment, hypertensive disorders in pregnancy and referral for delivery at HTH being predictors. Providing incentives to attract specialists to district hospitals could help in the effective management of hypertensive disorders in pregnancy and subsequently reduce the odds of preterm livebirths. The reproductive and child health unit of the Ghana Health Service is encouraged to come up with new and effective ways of sustaining the appreciable adequate ANC contacts and increasing IPTp uptake as these interventions have been shown to reduce odds of preterm births. Larger studies are needed to further examine the relation between employment and pregnancy outcomes.

## Supporting information

**S1 Dataset. Dataset underlying the study findings (sociodemographic characteristics).**
(XLSX)

**S2 Dataset. Second dataset underlying the study findings (inferential analysis and outcomes).**
(XLSX)

**S1 STROBE. Filled STROBE Statement.**
(DOCX)

## Acknowledgments

We are grateful to the study participants for their participation and the management of the Ho Teaching Hospital for giving us permission to conduct the study.

## Author contributions

**Conceptualization:** Anthony Kwame Dah, Joseph Osarfo.

**Data curation:** Anthony Kwame Dah, Joseph Osarfo, Hintermann Mbroh, Gifty Dufie Ampofo.

**Formal analysis:** Anthony Kwame Dah, Joseph Osarfo.

**Methodology:** Anthony Kwame Dah, Joseph Osarfo, Gifty Dufie Ampofo, Emmanuel Senanu Komla Morhe.

**Project administration:** Anthony Kwame Dah, Joseph Osarfo, Hintermann Mbroh, Adu Appiah-Kubi, Wisdom Klutse Azanu, Michael Amoh.

**Supervision:** Anthony Kwame Dah, Joseph Osarfo, Hintermann Mbroh, Adu Appiah-Kubi, Wisdom Klutse Azanu, Michael Amoh, Emmanuel Senanu Komla Morhe.

**Validation:** Michael Amoh.

**Visualization:** Joseph Osarfo.

**Writing – original draft:** Anthony Kwame Dah, Joseph Osarfo, Hintermann Mbroh, Gifty Dufie Ampofo.

**Writing – review & editing:** Anthony Kwame Dah, Joseph Osarfo, Hintermann Mbroh, Gifty Dufie Ampofo, Adu Appiah-Kubi, Wisdom Klutse Azanu, Emmanuel Senanu Komla Morhe.

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
