## [Decision Letter · Decision Letter 0]

3 Aug 2025

Dear Dr. Osarfo

We look forward to receiving your revised manuscript.

Kind regards,

Adaoha Pearl Agu, MBBS, MSc, FMCPH

Academic Editor

PLOS ONE

Journal Requirements:

Additional Editor Comments:

The article is well written, data supports the conclusions and the statistical analysis is appropriate. However, please address the following concerns in addition to those raised by the reviewer.

Study Procedures and Data Collection Tool: State at what point in the study the questionnaires were administered and how this was done. State your rationale for using 60% of your data in the quality control mentioned in line 137

Data Management and Analysis: State the type of Incidence measure you used i.e Incidence proportion.

Results : State if there was any loss to follow up as this may bias the outcome. State that the 119 admissions to NICU in line 240 were from your study population as I imagine that women who deliver in other hospitals may also have had their babies referred to NICU but they were not part of your study.

Carry out a grammar and spelling check throughout the article.

Reviewers' comments:

Reviewer's Responses to Questions

**Comments to the Author**

1. Is the manuscript technically sound, and do the data support the conclusions?

Reviewer #1: Yes

2. Has the statistical analysis been performed appropriately and rigorously?

Reviewer #1: Yes

3. Have the authors made all data underlying the findings in their manuscript fully available?

Reviewer #1: Yes

4. Is the manuscript presented in an intelligible fashion and written in standard English?

Reviewer #1: Yes

Reviewer #1: This is fairly well written article but would need to look at the following areas. The authors indicate that "no formal sample size was done". I guess they were referring to "total population sampling method."

This was a prospective cohort study but the authors indicate that birthweight, Apgar scores, congenital anomaly and NICU admissions were recorded at delivery or later extracted from the delivery register. This cast doubts on the reliability of the data. By implication, there was no trained team to carry out the work but rather the authors were relying on whatever were recorded in the case notes, individual differences notwithstanding. This did not even come up under their limitations.

The spelling of pyrimethamine should be corrected - line 128. The authors should stick to one term either hypertensive disorders of pregnancy or hypertensive disorders in pregnancy in the article.

Line 320: "These" should be corrected to "these".

Line 334: "International Federation of Gynaecology and Obstetrics" not "Obstetrician"

**Do you want your identity to be public for this peer review?** For information about this choice, including consent withdrawal, please see our Privacy Policy

Reviewer #1: No

---

## [Author Response · Author response to Decision Letter 1]

5 Aug 2025

Response to Additional Editor Comments

1. State at what point in the study the questionnaires were administered and how this was done

Response: This has been addressed under ‘Study Procedures and Data Collection Tool’. The following has been inserted from line 129 to line 137 in the revised manuscript;

“Specifically, the questionnaires were administered when the women were admitted for labour or at 1-2 hours after delivery when they were stable and at admission for pre-labour caesarean section. Where a woman was in severe labour pain at admission, questionnaire administration was delayed till after delivery when she was stable. The questionnaires were administered by three registered midwives trained over two days in the relevant data collection procedures. Their training also included translation of the questionnaire from English to the local Ewe and Twi languages for uniformity as described in an earlier study [21]. Questionnaire administration was done in English, Ewe or Twi.”

2. State your rationale for using 60% of your data in the quality control mentioned in line 137.

Response: This has been stated as shown below in lines 147-151 in the revised manuscript

“The choice of 60% was discretionary and involved cross-checking three out of every five questionnaires. The registered midwives used for data collection, by default, were already familiar with the study variables and were positioned to identify potential data discrepancies themselves. The choice of using 60% of data collected in quality control was thus deemed reasonable.”

3. Data Management and Analysis: State the type of Incidence measure you used i.e Incidence proportion.

Response: This has been stated under ‘Data Management and Analysis’

4. State if there was any loss to follow up as this may bias the outcome.

Response: There was no loss to follow up

5. State that the 119 admissions to NICU in line 240 were from your study population as I imagine that women who deliver in other hospitals may also have had their babies referred to NICU but they were not part of your study.

Response: This has been done in line 254

6. Carry out a grammar and spelling check throughout the article.

Response: This has been done

Responses to Comments from Reviewer 1

1. The authors indicate that "no formal sample size was done". I guess they were referring to "total population sampling method."

Response: Yes, we agree with the reviewer on this. The relevant section (Sample Size Determination) has been revised for clarity on this point and is shown below;

“No formal sample size calculation was done. All women admitted for labour and delivery or pre-labour caesarean sections from 1st October, 2019 to 31st March, 2020 and their newborns were included if found eligible but excluded those who had stillbirths or delivery before 28 weeks. Delivery before 28 weeks is considered an abortion in our settings. In all, 666 mothers and 680 live newborns from a total sampling of eligible participants over the defined period were included for data analysis.”

2. This was a prospective cohort study but the authors indicate that birthweight, Apgar scores, congenital anomaly and NICU admissions were recorded at delivery or later extracted from the delivery register. This cast doubts on the reliability of the data. By implication, there was no trained team to carry out the work but rather the authors were relying on whatever were recorded in the case notes, individual differences notwithstanding. This did not even come up under their limitations.

Response: It was a prospective cohort study as the reviewer rightly points out and was therefore not based on extraction of already-collected data from existing registers.

Primary data were collected from questionnaires administered (by trained midwives) at admission for labour, at 1-2 hours after delivery when the woman was stable and at admission for pre-labour caesarean section (see lines 129-137).

In a few cases where either the mother or the baby was unstable and decisive clinical action had to be taken, it was not deemed prudent to collect data immediately as we would have done for a stable patient. We therefore waited for some hours (up to 24-48 hours) and went back to capture relevant data from appropriate registers into which the ward staff would have already entered such data. The ward staff in question were registered midwives as well.

We assure the reviewer that the data on which the manuscript is based is reliable primary data and not retrospective data. The authors opine that there is no limitation in relation to this matter that needs to be addressed.

3. The spelling of pyrimethamine should be corrected - line 128

Response: This has been done in line 129 now

4. The authors should stick to one term either hypertensive disorders of pregnancy or hypertensive disorders in pregnancy in the article.

Responses: ‘hypertensive disorders in pregnancy’ is used in the manuscript. All corrections have been done (lines 171 and 229 and the footnote under Table 2)

5. Line 320: "These" should be corrected to "these".

Response: The correction has been done in line 336 now

6. Line 334: "International Federation of Gynaecology and Obstetrics" not "Obstetrician"

Response: the correction has been done in line 350 now

---

## [Editor Report · Decision Letter 1]

13 Aug 2025

Incidence, risk factors and outcomes of preterm live births in a tertiary health facility in Ghana

PONE-D-24-30104R1

Dear Dr. Osarfo,

We’re pleased to inform you that your manuscript has been judged scientifically suitable for publication and will be formally accepted for publication once it meets all outstanding technical requirements.

Kind regards,

Adaoha Pearl Agu, MBBS, MSc, FMCPH

Academic Editor

PLOS ONE
---

## [Editor Report · Acceptance letter]

PONE-D-24-30104R1

PLOS ONE

Dear Dr. Osarfo,

I'm pleased to inform you that your manuscript has been deemed suitable for publication in PLOS ONE. Congratulations! Your manuscript is now being handed over to our production team.

Kind regards,

on behalf of

Dr. Adaoha Pearl Agu

Academic Editor

PLOS ONE